# Bayesian Optimization of Risk Measures

**Sait Cakmak**
Georgia Institute of Technology
scakmak3@gatech.edu

**Raul Astudillo**
Cornell University
ra598@cornell.edu

**Peter Frazier**
Cornell University
pf98@cornell.edu

**Enlu Zhou**
Georgia Institute of Technology
enlu.zhou@isye.gatech.edu

## Abstract

We consider Bayesian optimization of objective functions of the form $\rho[F(x, W)]$, where $F$ is a black-box expensive-to-evaluate function and $\rho$ denotes either the VaR or CVaR risk measure, computed with respect to the randomness induced by the environmental random variable $W$. Such problems arise in decision making under uncertainty, such as in portfolio optimization and robust systems design. We propose a family of novel Bayesian optimization algorithms that exploit the structure of the objective function to substantially improve sampling efficiency. Instead of modeling the objective function directly as is typical in Bayesian optimization, these algorithms model $F$ as a Gaussian process, and use the implied posterior on the objective function to decide which points to evaluate. We demonstrate the effectiveness of our approach in a variety of numerical experiments.

## 1 Introduction

Traditional Bayesian optimization (BO) has focused on problems of the form $\min_x F(x)$, or more generally $\min_x \mathbb{E}[F(x, W)]$, where $F$ is a time-consuming black box function that does not provide derivatives, and $W$ is a random variable. This has seen an enormous impact, and has expanded from hyper-parameter tuning [1, 2] to more sophisticated applications such as drug discovery and robot locomotion [3, 4, 5, 6]. However, in many truly high-stakes settings, optimizing average performance is inappropriate; we must be risk-averse. In such settings, risk measures have become a crucial tool for quantifying risk. For instance, by law, banks are regulated using the Value-at-Risk (VaR) [7]. Risk measures have also been used in cancer treatment planning [8, 9], healthcare operations [10], natural resource management [11], disaster management [12], data-driven stochastic optimization [13], and risk quantification in stochastic simulation [14].

In this work, we consider risk averse optimization of the form $\min_x \rho[F(x, W)]$, where $\rho$ is a *risk measure* that maps the probability distribution of $F(x, W)$ (induced by the randomness on $W$) onto a real number describing its level of risk. We focus on the setting, where, during the evaluation stage, $F$ can be evaluated for any $(x, w) \in \mathcal{X} \times \mathcal{W}$, e.g., using a simulation oracle. After the evaluation stage, we choose a decision $x^*$ to be implemented in the real world, nature chooses the random $W$, and the objective $F(x^*, W)$ is then realized.

When $F$ is inexpensive and has convenient analytic structure, optimizing against a risk measure is well understood [15, 16, 17, 18]. However, when $F$ is *expensive-to-evaluate*, *derivative-free*, or is a *black box*, the existing literature is inadequate in answering the problem. A naive approach would be to use the standard BO algorithms with observations of $\rho[F(x, W)]$. However, a single evaluation of the objective function, $\rho[F(\cdot, W)]$, requires multiple evaluations of $F$, which can be prohibitively expensive when the evaluations of $F$ are expensive. For example, if each evaluation

of $F$ takes one hour, and we need $10^2$ samples to obtain a high-accuracy estimate of $\rho[F(x, W)]$, a single evaluation of the objective function would require more than four days. Moreover, if we do not obtain a high-accuracy estimate, estimators are typically biased [19] in a way that is unaccounted for by traditional Gaussian process regression. For the risk measure Value-at-Risk, the recent work of [20] addresses this issue by modeling $\text{VaR}_\alpha[F(x, W)]$ using individual observations of $F(x, w)$. However, their method only chooses the $x$ to evaluate, and $w$ is randomly set according to its environmental distribution. As evidenced by [21, 22], jointly selecting $x$ and $w$ to evaluate offers significant improvements to query efficiency. This ability, unlocked when evaluations are made using a simulation oracle, is leveraged by the methods we introduce here.

As an example of the benefits of choosing $w$ intelligently, consider a function $F$ that is monotone in $w$. To optimize VaR of $F$, we only need to evaluate a single value of $w$, the one that corresponds to the VaR, and evaluating any other $w$ is simply inefficient. In a black-box setting, we would not typically know that $F$ was monotone but, by modeling $F$ directly, we can discover the regions of $w$ that matter and focus most of our effort into selectively evaluating $w$ from these regions.

In this paper, we focus on two risk measures commonly used in practice, Value-at-Risk (VaR) and Conditional Value-at-Risk (CVaR); and develop a novel approach that overcomes the aforementioned challenges. Our contributions are summarized as follows:

- To the best of our knowledge, our work is the first to consider BO of risk measures while leveraging the ability to choose $x$ *and* $w$ at query time. The selection of $w$ enables efficient search of the solution space, and is a significant contributor to the success of our algorithms.

- We provide a novel one-step Bayes optimal algorithm $\rho$KG, and a fast approximation $\rho$KG$^{apx}$ that performs well in numerical experiments; significantly improving the sampling efficiency over the state-of-the-art BO methods.

- We combine ideas from different strands of literature to derive gradient estimators for efficient optimization of $\rho$KG and $\rho$KG$^{apx}$, which are shown to be asymptotically unbiased and consistent.

- To further improve the computational efficiency, we propose a two time scale optimization approach that is broadly applicable for optimizing acquisition functions whose computation involves solving an inner optimization problem.

The remainder of this paper is organized as follows. Section 2 provides a brief background on BO and risk measures. Section 3 formally introduces the problem setting. Section 4 introduces the statistical model on $F$, a Gaussian process (GP) prior, and explains how to estimate VaR and CVaR of a GP. Section 5 introduces $\rho$KG, a knowledge gradient type of acquisition function for optimization of VaR and CVaR, along with a cheaper approximation, $\rho$KG$^{apx}$, and efficient optimization of both acquisition functions. Section 6 presents numerical experiments demonstrating the performance of the algorithms developed here. Finally, the paper is concluded in Section 7.

## 2 Background

### 2.1 Bayesian optimization

BO is a framework for global optimization of expensive-to-evaluate, black-box objective functions [23], whose origins date back to the seminal work of [24]. Among the various types of acquisition functions in BO [25, 26, 27], our proposed acquisition functions can be catalogued as knowledge gradient (KG) methods, which have been key to extending BO beyond the classical setting, allowing for parallel evaluations [28], multi-fidelity observations [29], and gradient observations [30].

Within the BO literature, a closely related work to ours is [22], which studies the optimization of $\mathbb{E}_W[F(x, W)]$, while jointly selecting $x$ and $w$ to evaluate. Due to linearity of the expectation, the GP prior on $F(x, w)$ translates into a GP prior on $\mathbb{E}_W[F(x, W)]$, a fact that is leveraged to efficiently compute the one-step Bayes optimal policy, also known as the KG policy. Although it is not the focus of this study, since CVaR at risk level $\alpha = 0$ is the expectation, our methods can be directly applied for solving the expectation problem, and the resulting algorithm is equivalent to the algorithm proposed in [22].

Our work also falls within a strand of the BO literature that aims to find solutions that are risk-averse to the effect of an unknown environmental variable [31, 32, 20, 33, 34]. Worst-case optimization, also known as minimax or robust optimization, is considered by [31] and [32], whereas [33] and [34] consider distributionally robust optimization [35]. Within this line of research, [20], which studies the optimization of $\text{VaR}_\alpha[F(x, W)]$, is arguably the most closely related to work. Like our statistical model, their model is also built using individual observations of $F(x, w)$ and not directly using observations of $\text{VaR}_\alpha[F(x, W)]$. However, unlike our approach, their approach is only able to choose at which $x$ to evaluate. Our approach jointly chooses $x$ and $w$ to evaluate, which is critical when $\mathcal{W}$ is large. Moreover, our model allows for noisy evaluations of $F(x, w)$, which could introduce additional bias in their model.

## 2.2 Risk measures

We recall that a risk measure (cf. [36, 37, 38, 39]) is a functional that maps probability distributions onto a real number. Risk measures offer a middle ground between the risk-neutral expectation operator and the worst-case performance measure, which is often more interpretable than expected utility [13]. For a generic random variable $Z$, we often use the notation $\rho[Z]$ to indicate the risk measure evaluated on the *distribution* of $Z$.

The most widely used risk measure is VaR [40], which measures the maximum possible loss after excluding worst outcomes with a total probability of $1 - \alpha$, and is defined as $\text{VaR}_\alpha[Z] = \inf\{t : P_Z(Z \leq t) \geq \alpha\}$, where $P_Z$ indicates the distribution of $Z$. Another widely used risk measure is CVaR, which is the expectation of the worst losses with a total probability of $1 - \alpha$, and is given by $\text{CVaR}_\alpha[Z] = \mathbb{E}_Z[Z \mid Z \geq \text{VaR}_\alpha(Z)]$.

VaR and CVaR have been applied in a wide range of settings, including simulation optimization under input uncertainty [13, 18], insurance [41] and risk management [42]. They are widely used in finance, and VaR is encoded in the Basel II accord [43]. We refer the reader to [44] for a broader discussion on VaR and CVaR.

## 3 Problem setup

We consider the optimization problem

$$\min_{x \in \mathcal{X}} \rho\left[F(x, W)\right], \tag{1}$$

where $\mathcal{X} \subset \mathbb{R}^{d_\mathcal{X}}$ is a simple compact set, e.g., a hyper-rectangle; $W$ is a random variable with probability distribution $\mathbb{P}_W$ and compact support $\mathcal{W} \subset \mathbb{R}^{d_\mathcal{W}}$; and $\rho$ is a known risk measure, mapping the random variable $F(x, W)$ (induced by $W$) to a real number. We assume that $F : \mathcal{X} \times \mathcal{W} \to \mathbb{R}$ is a continuous black-box function whose evaluations are expensive, i.e., each evaluation takes hours or days, has significant monetary cost, or the number of evaluations is limited for some other reason. We also assume that evaluations of $F$ are either noise-free or observed with independent normally distributed noise with known variance.

We emphasize that the risk measure $\rho$ is only over the randomness in $W$, and $\rho\left[F(x, W)\right]$ is calculated holding $F$ fixed. For example, if $\rho$ is VaR, and $W$ is a continuous random variable with density $p(w)$, then this is explicitly written as:

$$\text{VaR}_\alpha[F(x, W)] = \inf\left\{t : \int_\mathcal{W} \mathbb{1}\{F(x, w) \leq t\}p(w)\, dw \geq \alpha\right\} \tag{2}$$

Later, we will model $F$ as being drawn from a GP, but we will continue to compute $\rho\left[F(x, W)\right]$ only over the randomness in $W$. Thus, this quantity is a function of $F$ and $x$, and will be random with a distribution induced by the distribution on $F$ (and more precisely by the distribution on $F(x, \cdot)$).

## 4 Statistical model

We assume a level of familiarity with GPs and refer the reader to [45] for details. We place a GP prior on $F$, specified by a mean function $\mu_0 : \mathcal{X} \times \mathcal{W} \to \mathbb{R}$ and a positive definite covariance function $\Sigma_0 : (\mathcal{X} \times \mathcal{W})^2 \to \mathbb{R}^+$, and assume that queries of $F$ are of the form $y_i = F(x_i, w_i) + \epsilon_i$, where

$\epsilon_i \sim N(0, \sigma^2)$ is independent across evaluations. The posterior distribution on $F$ given the history after $n$ evaluations, $\mathcal{F}_n := \{(x_i, w_i), y_i\}_{i=1}^n$, is again a GP with mean and covariance functions $\mu_n$ and $\Sigma_n$, which can be computed in closed form in terms of $\mu_0$ and $\Sigma_0$.

The GP posterior distribution on $F$ implies a posterior distribution on the mapping $x \mapsto \rho[F(x, W)]$. In contrast with the case where $\rho$ is the expectation operator, this distribution is, in general, non-Gaussian, rendering computations more challenging. As we discuss next, quantities of interest can still be computed following a simple Monte Carlo (MC) approach. Specifically, we discuss how to compute the posterior mean of $\rho[F(x, W)]$, i.e., $\mathbb{E}_n[\rho[F(x, W)]]$, where $\mathbb{E}_n$ denotes the conditional expectation given $\mathcal{F}_n$.

Our approach to estimating the value of $\mathbb{E}_n[\rho[F(x, W)]]$ builds on samples of $[F(x, w) : w \in \mathcal{W}]$ drawn from the posterior, and the corresponding implied values of $\rho[F(x, W)]$. For the purposes of this discussion, we assume $\mathcal{W}$ is finite and small, say $\mathcal{W} = \{w_1, \ldots, w_L\}$, and that $\mathbb{P}_W$ is uniform over $\mathcal{W}$. When the cardinality of $\mathcal{W}$ is finite but large or $\mathbb{P}_W$ is continuous, we instead use a set of $L$ i.i.d. samples drawn from $\mathbb{P}_W$.

Denote $w_{1:L} = [w_1, \ldots, w_L]$ and $F(x, w_{1:L}) = [F(x, w_1), \ldots, F(x, w_L)]$. The time-$n$ joint posterior distribution of $F(x, w_{1:L})$ is normal with mean vector $\mu_n(x, w_{1:L})$ and covariance matrix $\Sigma_n(x, w_{1:L}, x, w_{1:L})$. A sample from this distribution can be obtained via the reparameterization trick [46], i.e., as $\mu_n(x, w_{1:L}) + C_n(x, w_{1:L})Z$, where is $C_n(x, w_{1:L})$ is the Cholesky factor of $\Sigma_n(x, w_{1:L}, x, w_{1:L})$ and $Z$ is drawn from the $L$-variate standard normal distribution.

Let $\widehat{F}(x, w_{1:L})$ denote a realization of $F(x, w_{1:L})$, computed as described above. The corresponding MC estimate of $\rho[F(x, W)]$ can be calculated as the empirical risk measure corresponding to this realization, which, under the assumption that $\mathbb{P}_{\mathcal{W}}$ is uniform over $\mathcal{W}$, can obtained by ordering the coordinates of $\widehat{F}(x, w_{1:L})$ so that $\widehat{F}(x, w_{(1)}) \le \widehat{F}(x, w_{(2)}) \le \ldots \le \widehat{F}(x, w_{(L)})$, and letting

$$\widehat{v}(x) := \widehat{F}(x, w_{(\lceil L\alpha \rceil)}) \quad \text{and} \quad \widehat{c}(x) := \frac{1}{\lceil L(1-\alpha) \rceil} \sum_{j=\lceil L\alpha \rceil}^{L} \widehat{F}(x, w_{(j)}), \tag{3}$$

be the empirical VaR and CVaR respectively (cf. [44]), where $\lceil \cdot \rceil$ is the ceiling operator. This can be easily extended to non-uniform discrete $\mathbb{P}_W$ by accounting for the probability mass of each $w \in \mathcal{W}$.

Finally, if $\widehat{v}^j(x)$ and $\widehat{c}^j(x)$, $j = 1, \ldots, M$, are samples of $\text{VaR}_\alpha[F(x, W)]$ and $\text{CVaR}_\alpha[F(x, W)]$, obtained as described above. Then, MC estimates of $\mathbb{E}_n[\text{VaR}_\alpha[F(x, W)]]$ and $\mathbb{E}_n[\text{CVaR}_\alpha[F(x, W)]]$ are given by $\frac{1}{M} \sum_{j=1}^M \widehat{v}^j(x)$ and $\frac{1}{M} \sum_{j=1}^M \widehat{c}^j(x)$, respectively.

## 5 The $\rho$KG acquisition function

As is standard in the BO literature, our algorithm's search is guided by an acquisition function, whose maximization indicates the next point to evaluate. Our proposed acquisition function generalizes the well-known knowledge gradient acquisition function [26], which has been generalized to other settings such as parallel and multi-fidelity optimization [28, 29].

Before formally introducing our acquisition function, we note that, in our setting, choosing a decision $x$ to be implemented after the evaluation stage is complete, is not a straightforward task. If the cardinality of $\mathcal{W}$ is large, then the chances are $\rho[F(x, W)]$ will not be known exactly, as this would require evaluating $F(x, w)$ for all $w \in \mathcal{W}$. A common approach in such scenarios is to choose the decision with best expected objective value according to the posterior distribution (c.f. [23]),

$$\min_{x \in \mathcal{X}} \mathbb{E}_N[\rho[F(x, W)]]. \tag{4}$$

Having defined the choice of the decision $x$ to be implemented after the evaluation stage is complete, we can now introduce our acquisition function, the knowledge gradient for risk measures. We motivate this acquisition function by noting that, if we had to choose a decision with the information available at time $n$, the expected objective value we would get according to equation (4) is simply $\rho_n^* := \min_{x \in \mathcal{X}} \mathbb{E}_n[\rho[F(x, W)]]$. On the other hand, if we were allowed to make one additional evaluation, the expected objective value we would get is $\rho_{n+1}^* := \min_{x \in \mathcal{X}} \mathbb{E}_{n+1}[\rho[F(x, W)]]$. Therefore, $\rho_n^* - \rho_{n+1}^*$ measures the improvement due to making one additional evaluation.

We emphasize that, given the information available at time $n$, $\rho_{n+1}^*$ is random due to its dependence on the yet unobserved $(n+1)$-st evaluation. The knowledge gradient for risk measures acquisition

function is defined as the expected value of $\rho_n^* - \rho_{n+1}^*$ given the information available at time $n$ and the next point $(x, w)$ to evaluate, termed the "candidate" from here on:

$$\rho \text{KG}_n(x, w) = \mathbb{E}_n \left[ \rho_n^* - \rho_{n+1}^* \mid (x_{n+1}, w_{n+1}) = (x, w) \right] \qquad (5)$$

Our algorithm sequentially chooses the next point to evaluate as the candidate that maximizes (5), and is, by construction, one-step Bayes optimal.

## 5.1 Optimization of $\rho$KG

In this section, we discuss how to evaluate and optimize $\rho$KG using a sample average approximation (SAA) approach [47]. In a nutshell, this approach works by constructing an MC approximation of $\rho \text{KG}_n(x, w)$ that is deterministic given a finite set of *base samples* not depending on the candidate $(x, w)$. Such an approximation can be optimized using deterministic optimization methods. This is usually faster than optimizing the original acquisition function with stochastic optimization techniques. Below, we discuss how to construct this SAA. Moreover, we show that its gradients can be readily computed, thus allowing the use of higher-order optimization methods. A more detailed discussion of our approach to optimize $\rho$KG can be found in the supplement.

We begin by noting that $\rho_n^*$ does not depend on the candidate being evaluated. Therefore, maximizing $\rho$KG is equivalent to solving

$$\max_{(x, w) \in \mathcal{X} \times \mathcal{W}} \mathbb{E}_n \left[ -\rho_{n+1}^* \mid (x_{n+1}, w_{n+1}) = (x, w) \right]. \qquad (6)$$

The first step in building an SAA of (6) is to draw $K$ *fantasy* samples from the time-$n$ posterior distribution on $y_{n+1}$, which, conditioned on $(x_{n+1}, w_{n+1}) = (x, w)$, is Gaussian with mean $\mu_n(x, w)$ and variance $\Sigma_n(x, w, x, w)$. Using the reparameterization trick, these samples can be obtained as $\mu_n(x, w) + \sqrt{\Sigma_n(x, w, x, w)} Z^i$, $i = 1, \ldots, K$, where the *base samples* $Z^1, \ldots, Z^K$ are drawn from a standard normal distribution. These samples give rise to $K$ *fantasy* GP models of the posterior distribution at time $n + 1$, obtained by conditioning the GP model on the event $(x_{n+1}, w_{n+1}) = (x, w)$ and $y_{n+1} = \mu_n(x, w) + \sqrt{\Sigma_n(x, w, x, w)} Z^i$, i.e., by adding $\{(x, w), \mu_n(x, w) + \sqrt{\Sigma_n(x, w, x, w)} Z^i\}$ as the hypothetical $(n + 1)$-st observation.

For each fantasy GP model $i$, an MC estimate of $\mathbb{E}_{n+1}[\rho[F(x^i, W)]]$, $x^i \in \mathcal{X}$, can be constructed by averaging samples as described in Section 4. Let $\mathfrak{r}^{ij}(x^i)$ denote the $j$-th such sample corresponding to the $i$-th fantasy GP model, where the dependence of $\mathfrak{r}^{ij}(x^i)$ on the candidate $(x, w)$ and $Z^i$ is made implicit. Additional base samples needed to define $\mathfrak{r}^{ij}(x^i)$ are generated once and held fixed, so that it becomes a deterministic function of $x^i$ and $(x, w)$. The SAA of (6) is then given by

$$\max_{(x, w) \in \mathcal{X} \times \mathcal{W}} -\frac{1}{K} \sum_{i=1}^{K} \min_{x^i \in \mathcal{X}} \frac{1}{M} \sum_{j=1}^{M} \mathfrak{r}^{ij}(x^i). \qquad (7)$$

In the supplement, we show that the gradient of $\mathfrak{r}^{ij}$ with respect to $x^i$, denoted $\nabla_{x^i} \mathfrak{r}^{ij}(x^i)$, can be computed explicitly as $\nabla_{x^i} \widehat{F}^{ij}(x^i, w_{(\lceil L\alpha \rceil)})$ and $\frac{1}{\lceil L(1-\alpha) \rceil} \sum_{j=\lceil L\alpha \rceil}^{L} \nabla_{x^i} \widehat{F}^{ij}(x^i, w_{(j)})$ for VaR and CVaR respectively, where this notation is defined explicitly in the supplement. We use these gradients within the L-BFGS algorithm [48] to solve the inner optimization problems in (7). Moreover, we show that, under mild regularity conditions, the envelope theorem ([49], Corollary 4) can be used to express the gradient of the objective in (7) as

$$-\frac{1}{K} \sum_{i=1}^{K} \frac{1}{M} \sum_{j=1}^{M} \nabla_{(x, w)} \mathfrak{r}^{ij}(x_*^i), \qquad (8)$$

where $\nabla_{(x, w)} \mathfrak{r}^{ij}(x_*^i)$ denotes the gradient of $\mathfrak{r}^{ij}$ with respect to $(x, w)$ evaluated at $x_*^i$, and $x_*^i$ is the solution to the $i$-th inner optimization problem in (7). Again, we use these gradients within L-BFGS to solve (7). In addition, we also show that the above gradients are asymptotically unbiased and consistent gradient estimators as $K, L, M \to \infty$. Therefore, $\rho$KG can be maximized using (multi-start) stochastic gradient ascent (SGA), following an approach similar to a proposal in [30].

**Proposition 1.** *Under suitable regularity conditions, (8) is an asymptotically unbiased and consistent estimator of the gradient of $\rho$KG as $K, L, M \to \infty$.*

A formal statement of the proposition and its proof is given in the supplement for the case of $d^{\mathcal{W}} = 1$. It is also shown that selecting the $n$-th candidate to evaluate using the $\rho$KG algorithm, including the training of the GP model, has a computational complexity of $\mathcal{O}(Q_1 n^3 + Q_2 Q_3 KL[n^2 + Ln + L^2 + ML])$, where $Q_1, Q_2, Q_3$ are the number of L-BFGS [48] iterations performed for training the GP model, and the outer and inner optimization loops respectively.

**Remark 1.** *This and the preceding sections are explained using MC estimators. The same approach works using quasi-MC estimators, obtained by generating $Z$ in reparameterization (see Section 4) using Sobol sequences [50]. In practice we use quasi-MC because it improves computationally over a simple MC approach.*

## 5.2 The $\rho$KG$^{apx}$ approximation

The $\rho$KG algorithm is computationally intensive, as it requires solving a nested non-convex optimization problem. In a wide range of settings, the sampling efficiency it provides justifies its computational cost, but in certain not-so-expensive settings, a faster algorithm is desirable.

Inspired by the EI [25] and KGCP [26] acquisition functions, we propose $\rho$KG$^{apx}$, which replaces the inner optimization problem of $\rho$KG with a much simpler one. In $\rho$KG$^{apx}$, the inner optimization is restricted to the points $x$ that have been evaluated for at least one $w$, denoted by $\widetilde{\mathcal{X}}_n = \{x_{1:n}\}$, and the resulting value of the optimization problem is $\widetilde{\rho}_n^* = \min_{x \in \widetilde{\mathcal{X}}_n} \mathbb{E}_n[\rho[F(x, W)]]$. The intuition behind this is that the GP model is an extrapolation of the data. Thus, these points carry an immense amount of information on the GP model and the posterior objective, which makes them an ideal set of candidates to consider. The resulting approximation to $\rho$KG is,

$$\rho\text{KG}^{apx}(x, w) = \mathbb{E}_n\left[\widetilde{\rho}_n^* - \widetilde{\rho}_{n+1}^* \mid (x_{n+1}, w_{n+1}) = (x, w)\right]. \tag{9}$$

We note that, like $\rho$KG, $\rho$KG$^{apx}$ has an appealing one-step Bayes optimal interpretation; the maximizer of $\rho$KG$^{apx}$ is the one-step optimal point to evaluate if we restrict the choice of the decision to be implemented, $x$, among those that have been evaluated for at least one environmental condition, $w$.

## 5.3 Two time scale optimization

In this paper, we introduce two acquisition functions, $\rho$KG and $\rho$KG$^{apx}$. These acquisition functions share a nested structure, making their maximization computationally challenging. Here, we describe a novel two time scale optimization approach for reducing this computational burden.

Since the posterior mean and kernel are continuous functions of the data, if we fix the base samples used to generate the fantasy model and the GP sample path, a small perturbation to the candidate solution $(x, w)$ results in only a slight shift to the sample path. Thus, the optimal solutions to the inner problems, obtained using the previous candidate solution, should remain within a small neighborhood of a current high quality local optimal solution (and likely of the global solution). We can thus use the inner solutions obtained in the previous iteration to obtain a good approximation of the acquisition function value and its gradient for the current candidate. We utilize this observation by solving the inner optimization problem once every $T \approx 10$ iterations, and using this solution to evaluate $\rho$KG for the remaining $T - 1$ iterations. We refer to this approach as *two time scale optimization* and present an algorithmic description with more detail in the supplement.

The two time scale optimization approach outlined here is not limited to $\rho$KG, and can be applied to other acquisition functions that require nested optimization, such as [26, 51, 22]. In numerical testing, the two-time-scale optimization approach did not affect the performance of either of our algorithms while offering significant computational savings.

## 5.4 A visual analysis of the acquisition functions

Figure 1 plots a GP model based on 6 random samples taken from a 2-dim test function and the corresponding acquisition function values of $\rho$KG and $\rho$KG$^{apx}$ over $\mathcal{X} \times \mathcal{W}$. We use a CVaR objective with risk level $\alpha = 0.7$ and a uniform distribution over $\mathcal{W} = \{0/9, 1/9, \ldots, 9/9\}$.

First, observe that the $\rho$KG$^{apx}$ plot closely resembles $\rho$KG, supporting our claim that it is a good approximation. Second, the implied posterior on the objective shows a large uncertainty for larger values of $x$, and we expect a large $\rho$KG for these $x$ to encourage exploration. $\rho$KG plots show that

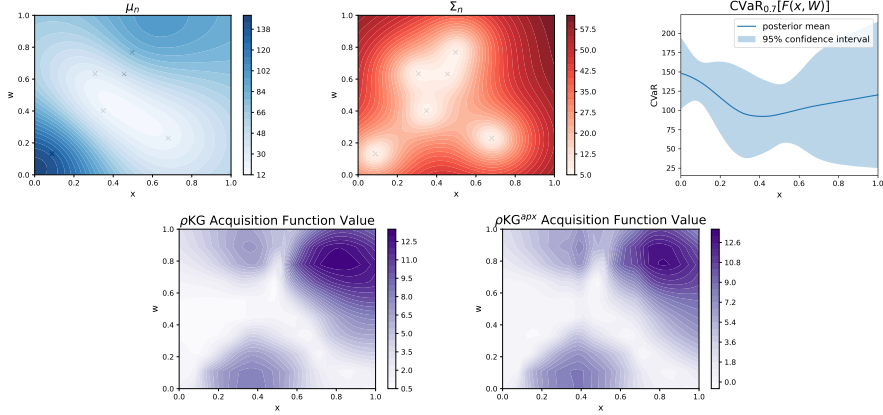

Figure 1: The top row shows the mean and variance functions of the posterior GP distribution on $F$, along with the implied (non-Gaussian) posterior distribution on $\text{CVaR}_{0.7}[F(\cdot, W)]$. The bottom row shows the $\rho\text{KG}$ and $\rho\text{KG}^{apx}$ acquisition functions implied by the above statistical model.

this is indeed the case, while also showing a large variation along the $w$ axis. In this region of $x \in [0.8, 1.0]$, the posterior uncertainty is much larger for larger values of $w$. These $w$ also happen to have a posterior mean near likely 0.7-quantiles of $F(x, \cdot)$, making them more informative about CVaR. Indeed, $\rho\text{KG}$ prefers these $w$, reaching a maximum near $w = 0.8$.

Another area of interest is the promising region of $x \in [0.2, 0.4]$, as it contains the minimizer of the current posterior mean of the objective. We observe both a posterior mean substantially below the likely 0.7-quantiles of $F(x, \cdot)$ and a low posterior uncertainty for $w \in [0.4, 0.6]$ in this region, making them less useful for estimating CVaR, and correspondingly low $\rho\text{KG}$. However, the $w$ at the two ends of the range are more likely to be near an 0.7-quantile and have larger uncertainty. Thus, an observation from these $(x, w)$ would help pinpoint the exact location of the minimizer of the posterior expectation of the objective and thus are associated with larger $\rho\text{KG}$.

We hope that these plots and accompanying discussion help the reader appreciate the value of considering the environmental variable, $w$, in designing the acquisition function.

## 6 Numerical experiments

In this section, we present several numerical examples that demonstrate the sampling efficiency of our algorithms. We compare our algorithms with the Expected Improvement (EI, [25]), Knowledge Gradient (KG, [51]), Upper Confidence Bound (UCB), and Max Value Entropy Search (MES, [52]) algorithms. We use the readily available implementations from the BoTorch package [53], and the default parameter values given there. The benchmark algorithms cannot utilize observations of $F(x, w)$ while optimizing $\rho[F(x, W)]$. Therefore, for these algorithms, we fit a GP on observations of $\rho[F(x, W)]$, which are obtained by evaluating a given $x \in \mathcal{X}$ for all $w \in \mathcal{W}$ (or a subset $\widetilde{\mathcal{W}}$) and then calculating the value of the risk measure on these samples. As a result, the benchmark algorithms require $|\mathcal{W}|$ (or $|\widetilde{\mathcal{W}}|$) samples per iteration whereas $\rho\text{KG}$ and $\rho\text{KG}^{apx}$ require only one. We could not compare with [20] since the code was not available at the time of the writing of this paper.

We optimize each acquisition function using the L-BFGS [48] algorithm with $10 \times (d^{\mathcal{X}} + d^{\mathcal{W}})$ restart points. The restart points are selected from $500 \times (d^{\mathcal{X}} + d^{\mathcal{W}})$ raw samples using a heuristic. For the inner optimization problem of $\rho\text{KG}$, we use $5 \times d^{\mathcal{X}}$ random restarts with $25 \times d^{\mathcal{X}}$ raw samples. For both $\rho\text{KG}$ and $\rho\text{KG}^{apx}$, we use the two time scale optimization where we solve the inner optimization problem once every 10 optimization iterations. $\rho\text{KG}$ and $\rho\text{KG}^{apx}$ are both estimated using $K = 10$ fantasy GP models, and $M = 40$ sample paths for each fantasy model.

We initialize each run of the benchmark algorithms with $2d^{\mathcal{X}} + 2$ starting points from the $\mathcal{X}$ space, and the corresponding evaluations of $\rho[F(x, W)]$ obtained by evaluating $F(x, w)$ for each $w \in \mathcal{W}$ (or $\widetilde{\mathcal{W}}$, to be specified for each problem). The GP models for $\rho\text{KG}$ and $\rho\text{KG}^{apx}$ are initialized using

the equivalent number of $F(x, w)$ evaluations, with $(x, w)$ randomly drawn from $\mathcal{X} \times \mathcal{W}$. Further details on experiment settings is given in the supplement. The code for our implementation of the algorithms and the experiments can be found at `https://github.com/saitcakmak/BoRisk`.

## 6.1 Synthetic test problems

The first two problems we consider are synthetic test functions from the BO literature. The first problem is the 4-dim Branin-Williams problem in [54]. We consider minimization of both VaR and CVaR at risk level $\alpha = 0.7$ with respect to the distribution of environmental variables $w = (x_2, x_3)$. The second problem we consider is the 7-dim $f_6(x_c, x_e)$ function from [31]. We formulate this problem for minimization of CVaR at risk level $\alpha = 0.75$ with respect to the distribution of the 3-dim environmental variable $x_e$. More details on these two problems can be found in the supplement.

## 6.2 Portfolio optimization problem

In this test problem, our goal is to tune the hyper-parameters of a trading strategy so as to maximize return under risk-aversion to random environmental conditions. We use CVXPortfolio [55] to simulate and optimize the evolution of a portfolio over a period of four years using open-source market data. Each evaluation of this simulator returns the average daily return over this period of time under the given combination of hyper-parameters and environmental conditions. The details of this simulator can be found in Sections 7.1-7.3 of [55].

The hyper-parameters to be optimized are the risk and trade aversion parameters, and the holding cost multiplier over the ranges $[0.1, 1000]$, $[5.5, 8.]$, and $[0.1, 100]$, respectively. The environmental variables are the bid-ask spread and the borrowing cost, which we assume are uniform over $[10^{-4}, 10^{-2}]$ and $[10^{-4}, 10^{-3}]$, respectively. For this problem, we use the VaR risk measure at risk level $\alpha = 0.8$ We use a random subset $\widetilde{\mathcal{W}}$ of size 40 for the inner computations of our algorithms, and a random subset of size 10 is used for the evaluations of $\text{VaR}_{0.8}[F(x, W)]$ by the benchmark algorithms.

Since this simulator is indeed expensive-to-evaluate, with each evaluation taking around 3 minutes, evaluating the performance of the various algorithms becomes prohibitively expensive. Therefore, in our experiments, we do not use the simulator directly. Instead, we build a surrogate function obtained as the mean function of a GP trained using evaluations of the actual simulator across 3000 points chosen according to a Sobol sampling design [50].

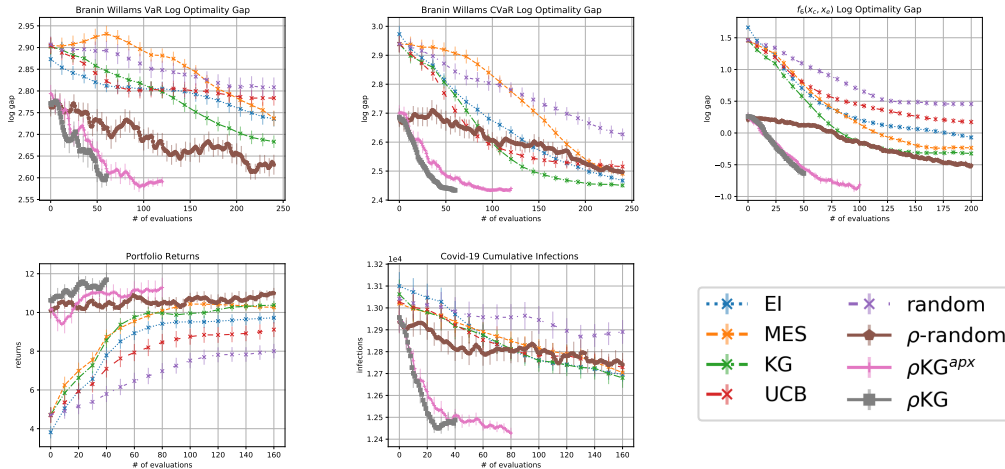

Figure 2: Top: The log optimality gap in Branin Williams with VaR (left), with CVaR (middle); and $f_6(x_c, x_e)$ (right). Bottom: The returns on Portfolio problem (left), the cumulative number of infections in the COVID-19 problem (middle) and the legend (right). The plots are plotted against the number of $F(x, w)$ evaluations, and are smoothed using a moving average of 3 iterations. Non-smoothed plots are given in the supplement.

### 6.3 Allocating COVID-19 testing capacity

In this example, we study allocation of a joint COVID-19 testing capacity between three neighboring populations (e.g., cities, counties). The objective is to allocate the testing capacity between the populations to minimize the total number of people infected with COVID-19 in a span of two weeks. We use the COVID-19 simulator provided in [56] and discussed in [57, 58]. The simulator models the interactions between individuals within the population, the disease spread and progression, testing and quarantining of positive cases. Contact tracing is performed for people who test positive, and the contacts that are identified are placed in quarantine.

We study three populations of sizes $5 \times 10^4$, $7.5 \times 10^4$, and $10^5$ that share a combined testing capacity of $10^4$ tests per day. The initial disease prevalence within each population is estimated to be in the range of $0.1 - 0.4\%$, $0.2 - 0.6\%$ and $0.2 - 0.8\%$ respectively. We assign a probability of $0.5$ to the middle of the range and $0.25$ to the two extremes, independently for each population. Thus, the initial prevalence within each population defines the environmental variables. We pick the fraction of testing capacity allocated to the first two populations as the decision variable (remaining capacity is allocated to the third), with the corresponding decision space $\mathcal{X} = \{x \in \mathbb{R}_+^2 : x_1 + x_2 \leq 1\}$. For the inner computations of $\rho\mathrm{KG}^{apx}$, we use the full $\mathcal{W}$ set, however, for the evaluations of benchmark algorithms we randomly sample a subset $\widetilde{\mathcal{W}}$ of size 10 to avoid using 27 evaluations per iteration.

### 6.4 Results

Figure 2 plots results of the experiments. Evaluations reported exclude the GP initialization, and the error bars denote one standard error. In each experiment, $\rho\mathrm{KG}$ and $\rho\mathrm{KG}^{apx}$ match or beat the performance of all benchmarks using less than half as many function evaluations, thus, demonstrating superior sampling efficiency. In the experiments $\widetilde{W}$ was intentionally kept small (between 8-12) to avoid giving our methods an outsized advantage. If $\widetilde{W}$ were larger, the benchmarks would use up even more evaluations per iteration, and our algorithms would provide an even larger benefit.

To demonstrate the benefit of using our novel statistical model, we included experiments with two random sampling strategies. The one labeled "random" evaluates $\rho[F(x, W)]$, and uses the corresponding GP model over $\mathcal{X}$. "$\rho$-random", on the other hand, evaluates $F(x, w)$ at a randomly selected $(x, w)$, uses our statistical model, and reports $\arg\min_x \mathbb{E}_n[\rho[F(x, W)]]$ as the solution. The ability to survey the whole $\mathcal{X} \times \mathcal{W}$ space gives "$\rho$-random" a significant boost. We see that, despite choosing evaluations randomly, it is highly competitive against all the benchmarks, and outperforms "random" by a significant margin. This demonstrates the added value of our statistical model, which captures all the information available in the data, and suggests an additional cheap-to-implement algorithm that is useful whenever $F(x, w)$ is cheap enough to render other algorithms too expensive.

The supplement presents additional plots comparing algorithm run-times. Data from Branin Williams and $f_6(x_c, x_w)$ show that even with only moderately expensive function evaluations (a few minutes per evaluation), our algorithms save time compared with the benchmarks presented here.

## 7 Conclusion

In this work, we introduced a novel Bayesian optimization approach for solving problems of the form $\min_x \rho[F(x, W)]$, where $\rho$ is a risk measure and $F$ is a black-box function that can be evaluated for any $(x, w) \in \mathcal{X} \times \mathcal{W}$. By modeling $F$ with a GP, instead of the objective function directly as is typical in Bayesian optimization, our approach is able to leverage more fine-grained information and, importantly, to jointly select both $x$ and $w$ at which to evaluate $F$. This allows our algorithms to significantly improve sampling efficiency over existing Bayesian optimization methods.

We propose two acquisition functions, $\rho\mathrm{KG}$, which is one step-Bayes optimal, and a principled cheap approximation, $\rho\mathrm{KG}^{apx}$, along with an efficient, gradient-based approach to optimize them. To further improve numerical efficiency, we introduced a two time scale optimization approach that is broadly applicable for acquisition functions that require a nested optimization.

## Broader Impact

Our work is of interest whenever one needs to make a decision guided by an expensive simulator, and subject to environmental uncertainties. Such scenarios arise in simulation assisted medical decision making [4], in financial risk management [7, 59, 60], in public policy, and disaster management [61].

The impact of our algorithms could be summarized as facilitating risk averse decision making. In many scenarios, risk averse approaches result in decisions that are more robust to environmental uncertainties compared to decisions resulting from common risk neutral alternatives. For example, in a financial crisis, an earlier risk-averse decision of holding more cash and other low-risk securities might prevent large losses or even the default of a financial institution. As another example, a risk averse decision of stockpiling of excess medical supplies in non-crisis times would alleviate the shortages faces during crisis times, such as the COVID-19 pandemic we are facing today.

On the negative side of things, the risk averse decisions we facilitate may not always benefit all stakeholders. For a commercial bank, a risk averse approach may suggest a higher credit score threshold for making loans, which might end up preventing certain groups from access to much needed credit.

In our case, the failure of the system would mean a poor optimization of the objective, and recommendation of a bad solution. Implementation of a bad decision can have harmful consequences in many settings; however, we imagine that any solution recommended by our algorithms would then be evaluated using the simulator, thus preventing the implementation of said decision.

Our methods do not rely on training data, and only require noisy evaluations of the function value. Thus, it can be said that our method does not leverage any bias in any training data. However, the solutions recommended by our algorithms are only good up to the supplied function evaluations, thus are directly affected by any biases built into the simulator used for these evaluations.

## Acknowledgments and Disclosure of Funding

The authors gratefully acknowledge the support by the National Science Foundation under Grants CAREER CMMI-1453934 and CCF-1740822; and the Air Force Office of Scientific Research under Grants FA9550-19-1-0283 and FA9550-15-1-0038. We also thank the anonymous reviewers, whose comments helped improve and clarify the presentation of our paper.

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
