[Reviews · NeurIPS 2020]

Review 1

Summary and Contributions: This manuscript considers the optimization of a risk measure (in particular either value-at-risk or conditional value-at-risk) applied to the distribution of an objective function value f(x, w) where w is an unknown environmental variable. Most treatments of Bayesian optimization consider optimizing the _expectation_ of f(x, w), but such an approach is inherently risk neutral. In scenarios were risk aversion (or I suppose risk seeking) is desired, there are not many options in the Bayesian optimization family. The authors propose a straightforward but apparently effective approach: - we define a utility function of data to be the global maximum of the expected risk measure conditioned on the data - we then create an acquisition function by computing the one-step lookahead marginal gain in this utility, using Monte Carlo estimation for the intractable expectations that arise and random restart for the nonconvex global optimization required This is already sufficient to specify a functional algorithm, but the authors provide some additional enhancements: - a Monte Carlo estimate of the gradient of the acquisition function to accelerate optimization - a simple approximation that avoids the global optimization required, replacing it with a finite approximation - a two time-scale optimization approach to accelerate the optimization of the acquisition function by only recomputing some expensive quantities every few iterations. These contributions are supported by a series of experiments and the proposed algorithm appears to be competitive. [post-rebuttal] I would like to thank the authors for their thoughtful and engaging response!

Strengths: I believe that risk-aware optimization is an important and understudied problem in the Bayesian optimization literature, and this work proposes a natural solution that appears to work well in practice. Every component of the algorithm is reasonable and is supported by recent computational developments to ensure it can be implemented effectively (in particular the approximation of the gradient). This work would be of interest to anyone from the Bayesian optimization community, which has high overlap with the NeurIPS community. All of the theoretical analysis that could be reasonably expected of a work like this has been performed satisfactorily (something like a regret bound would obviously be out of the question for a work of this scope and would entail a massive amount of work).

Weaknesses: I believe the major weakness of this work is one of novelty. Nearly all of the ideas that comprise the proposed algorithm have been proposed before and the authors simply (but tactfully) string them together. Namely, to run down the major contributions in this context: - Bayesian optimization of risk measures rather than the risk-neutral optimization of expectation. I think this is completely natural and have myself wondered why this hasn't seen more work. However, despite the authors' claim in lines 26, I do not believe this is novel to this work but would be happy to be corrected (see comments in "relation to prior work." - The use of a risk measure applied to the distribution of p(x, w) rather than expectaiton as a utility. Again, this does not appear to be novel, although defining this utility in terms of the global optimum is perhaps slightly novel here, although it's natural in light of the knowledge gradient. - The use of one-step lookahead to derive a policy. This is standard practice. - The use of Monte Carlo estimation to approximate the proposed acquisition function. This is standard practice and perhaps not even the best approach; see "additional feedback" below. - The use of Monte Carlo + the envelope theorem to approximate the gradient of the acquisition function. This is equivalent to the approach popularized by Wu, et al. NeurIPS 2017 (as acknowledged by the authors), and this section in particular is very reminiscent of that work. In my opinion a reader already familiar would gain no additional insight from the current presentation. - The approximation in §5.3. This is equivalent to the KGCP approach from Scott, et al. SIAM J on Optimization, 2011 (as acknowledged by the authors) - That leaves perhaps only the two time-scale optimization approach in §5.4 as novel this work. This idea is relatively straightforward but may be useful in other KG style contexts. Of course I understand that there is novelty in the combination of all of these ideas, which are numerous, clearly well understood by the authors, and non-trivial. Perhaps that is enough. I am looking forward to discussing this question with my fellow reviewers. Note that I think the writing style of the paper may be partially responsible for my general impression (see comments on "clarity" below), as the relatively straightforward and to-the-point style perhaps lessened the perceived impact of each sequential idea.

Correctness: Yes, I see no major issues with the methodology, either theoretical or empirical.

Clarity: The paper is well written but in my opinion somewhat perfunctory. All of the major points are covered but there's little in the way of high-level discussion or motivation wrt alternative ideas. I would have appreciated some more nuanced discussion wrt the approximate method KGCP style method vs the full method in particular, as the approximation offers a great deal of implementation simplification and appears to work well.

Relation to Prior Work: I would appreciate some comments on the following, which seem highly related although their policies differ: - Gong, et al. Quantile Stein Variational Gradient Descent for Batch Bayesian Optimization. ICML 2019 [considers Bayesian optimization for a quantile objective equivalent to value-at-risk] - Torossian, et al. Bayesian Quantile and Expectile Optimization. arXiv:2001.04833 [stat.ML] [considers Bayesian optimization of risk measures including VaR] These should probably be cited and discussed by the authors, and I would appreciate some commentary in the context of the manuscript in the response.

Reproducibility: Yes

Additional Feedback: I have a few additional comments/questions that don't fit into the themes above: - line 77: Is it actually important that the support of W be compact? That assumption threw me for a loop as it (a) doesn't seem to actually be needed anywhere and (b) seems to be tautological: can't I rewrite any distribution over a unit cube by an inverse CDF transform? - line 100: What is the dimension of W that you would normally consider in practice? If the dimension is fairly small (which would seem to be the case, at least to me) it would seem that something like QMC or even adaptive numerical integration would offer faster convergence than simple Monte Carlo. The immediate jump to simple Monte Carlo is a bit weird as it's basically the worst possible way to estimate an expectation unless literally nothing else can work. - (6): I have a similar question here; why not use Gauss-Hermite quadrature for integrating over f? - line 153: The Wu, et al. paper also used this phrase "mild regularity," sweeping the exact condition under the rug. Can we provide something more explicit? In particular, isn't continuity enough? - line 161: I'm a bit confused why LBFGS would be used here considering that the gradient is only approximated/observed with noise; wouldn't some stochastic procedure be a better choice? - line 186: I was a bit confused here and may be reading the text wrong. There can certainly be massive discontinuities in the location of the global maximum of E[ρ] depending on the observed value (i.e., the maximum stays in its old location unless a sufficiently surprising value is seen, in which case it shifts suddenly to the neighborhood of the newest observation). - line 194: something weird happened here - references: there are some typos here, e.g. "bayesian" I suggest a once-over. - I have some additional minor comments regarding the content (typos, etc.) that I will include in a final version of the review if the work is headed towards acceptance.


Review 2

Summary and Contributions: The authors present a fully Bayesian approach to Bayesian optimization, where the Value at Risk is the basic quantity being optimized, and this quantity is indexed by a covariate and a Gaussian process. The overall quantity to optimize is the posterior mean (under observations from the GP) of the VAR. The acquisition function takes the form of a one-step change in this quantity as a single fictitious data point is incorporated in the posterior, and (brute-force) Monte Carlo techniques are used to approximate this acquisition function.

Strengths: The approach brings together essentially all of the ingredients of a real-world approach to Bayesian optimization, including the use of VAR, black-box evaluators, GPs, and reasonable approximations in the numerical optimization. While the full-fledged nature of this approach limits its applications to modest scale problems, it nonetheless sets a reasonably high bar for alternative strategies. The experimental evaluation is informative.

Weaknesses: There is limited novelty, but this isn't surprising given the mature nature of this field, and no theoretical analysis. The only theoretical statement has to do with asymptotically unbiased and consistent estimators of gradients, which isn't saying much in high-dimensional, small data domains such as the ones targeted here.

Correctness: Yes.

Clarity: Yes.

Relation to Prior Work: Yes.

Reproducibility: Yes

Additional Feedback: P(w) is bad notation for a distribution. typo: "over he randomness" In Eq (5), it's unappealing to be using "x" both as a dummy variable (in the min's on the RHS) and as a real variable (on the LHS). There's a formatting problem at the bottom of page 5 that prevented me from obtaining a full understanding of the implications of the two time scale approach. "Certain w in this are both have a small uncertainty..."


Review 3

Summary and Contributions: Most works on Bayesian optimization (BO) focus on the minimization w.r.t x of a function of the form E[F(x,W)] where W is a random variable and F a black-box function. However, in many applications, one does not minimize an expectation, but a measure of risk: for example the VaR in finance. Thus, this paper studies the minimization of rho[F(x,W)] where rho is a risk measure. Following a standard approach in Bayesian optimization, the authors model F by a Gaussian Process (GP), and the problem boils down to the choice of the acquisition function: given n observations, and the posterior distribution on F_n, which leads in turn to a distribution on rho[F_n(x,W)], at what point x_{n+1} should we compute F? The authors propose the rho-KG acquisition function. It is defined by the maximization of the improvement between the expected minimum of rho[F_{n}(x,W)] and rho[F_{n+1}(x,W)]. They then propose a Monte-Carlo strategy to estimate the gradient of this criterion, which allows to optimize it. However, the resulting strategy is too slow. Thus, they propose another acquisition function, which approximates the previous one. They compare the performances of these strategies to previous ones on synthetic examples, as well as on risk minimization in portfolio management and in Covid testing, with very promising results.

Strengths: 1) the extension of Bayesian optimization to popular, and important, measures of risk. 2) the good numerical performances of the proposed strategy.

Weaknesses: 1) the strategy is not clearly explained. The details on the implementation of the strategy, in Subsections 5.1 and 5.2, are extremely difficult to follow for a non-specialist of this literature, because no explanations on the quantities introduced are provided, and some formal definitions are missing. This looks almost like an informal description of the method. Consider for example the sentence "A fantasy GP model is the GP model obtained by conditioning the GP model on a fantasy observation simulated using the distribution implied by the GP model at the candidate point", it seems to me that a formal statement could make this sentence easier to understand. 2) it is difficult to assess the contribution of the authors, in the sense that the estimators of the gradient in the rho-KG strategy were studied in [42] and [43]...

Correctness: The exact statement of Proposition 1, that ensures the consistency of the estimates of the gradients, is given in the appendix; the proof seems to be correct. It's more difficult to assess the correctness of the experimental results, as the code is not provided by the authors. Note: the paper says that the code is online, but the link to the author website was anonymized. So, it is not possible to access this code. This code should have been included in the supplementary material.

Clarity: As written above, the explanation of the strategy should be improved (Subsections 5.1 and 5.2 especially).

Relation to Prior Work: List of references: fair presentation of previous works.

Reproducibility: No

Additional Feedback: None. ************************** ************************** I thank the authors for their detailed reply, in particular regarding the novelty of the work. I still believe that parts of the paper would require some rewriting, but I will increase my score from 4 to 5.


Review 4

Summary and Contributions: This paper extends Bayesian optimization to optimize a risk measure of a function w.r.t. an envinronmental variable, instead of optimizing the expected function value as in standard BO. The authors adopt the well-known risk measures of VaR and CVaR, and propose a novel and intuitive acquisition function based on knowldge gradient (KG). To overcome the computational cost of the proposed acquisition function, the authors adopt a series of approximation techniques. The resulting algorithm outperform a number of baselines in two real-world experiments for portfolio optimization and COVID-19 testing allocation.

Strengths: - The problem of optimizing risk measures of functions, instead of expected value of functions, using BO is very interesting and can certainly find applications in important areas. - Borrowing the two well-known risk measures from finance makes perfect sense. - The approximation techniques, although heuristic, seem to have good approximation quality. - The experiment on COVID-19 testing allocation is refreshing.

Weaknesses: Major: - I have a major concern regarding the setting you are considering: I think assuming that you can select the environmental variable w to query may not be realistic. Since W is an "environmental variable" and we want to be risk-averse w.r.t. the randomness it brings, I think it's usually uncontrolablle in practical applications. For example, in the COVID-19 experiment, I think in practice, we usually cannot choose the initial disease prevalence within each population, and we can only observe instead. I'm not familiar with portfolio optimization, but I assume the trade market environmental variables are also beyond our control. This makes me question the practical relevance of your approach. I think this is related to a deeper question as to whether it is reasonable to treat the environmental variable as part of the input space of GP. Also related to this point, in your experiments, the benchmark algorithms can only select x, while your algorithm can select both x and w. I'm afraid this might be an unfair advantage for your algorithm, hence bringing the question as to whether the experimental comparisons are fair. - I think there should be some experimental results showing that your algorithm (with approximation) is indeed computationally efficient. As the authors pointed out, their original acquisition function is computationally expensive, which is why they need a number of approximations. This naturally brings up concerns regarding whether these approximations indeed bring enough reduction in the computational cost such that your algorithm can converge faster than other benchmark algorithms in terms of computation time. This comparison can be done by either replacing the horizontal axis of Fig 3 with run time, or by adding additional results specifically comparing the run time of different algorithms. - The sampling-based techniques to approximate VaR and CVaR (Sec 5.1) seem to only hold for uniform distributions, which I think this can be very restrictive. Is it straightforward to extend to other more complicated distributions? I would need authors' explanations about the concerns above (especially the first one) in order for me to consider raising my score. Below are some less important points. - Sec 5.3, lines 179-180, what's the rationale behind "restricting the choice of the decision..... among those that have been evaluated...."? Why do you think this would be a good approximation. - Sec 5.5, I appreciate the authors' efforts in providing intuitions on how their algorithm works, but I find this section hard to understand. For example, I have trouble understanding the sentence on line 207, and the following few sentences... It would add large value to the paper if this section can be written more clearly. More Minor Points: - line 35: "state art" -> state-of-the-art - line 57: (1) is referred to here, but it hasn't been introduced yet - line 58: "operator," -> "operator." - line 68-70: it would be better if the authors could provide some interpretations/intuitions behind these two definitions of risk measures here - Equation (2), again, some interpretations behind this definnition would be nice - line 87: "he" -> the - line 120: "on" -> an - line 151-152, the expressions of the 2 gradients here look the same as the original expressions in lines 140-141, typo? - line 210: "are" -> area - I appreciate the lack of space, but I think a conclusion section is needed.

Correctness: I haven't read the detailed proof in the Supplement, but I think the technical details in the main text are correct.

Clarity: Overall it is well written, but there are a few places that can be improved: - It would greatly help the reader's understanding if some intuitive interprerations are provided alongside mathematical equations, e.g., the expressions of VaR and CVaR in Sec 2.2, Equation (2), etc. - The clarity of Sec 5.5, which can be a huge plus for the paper if written well, can be improved.

Relation to Prior Work: Related works are sufficiently covered. I think some of the most related works (e.g., [31] and [32]) need to be discussed with more detail.

Reproducibility: Yes

Additional Feedback: -----post-rebuttal update---- After reading the authors' rebuttal and discussing with other reviewers, I've decided to increase my score, since the authors addressed my previous concerns well, and I don't have other major concerns left. I would like the authors to make it explicit in the revised version that their algorithm requires a simulation, and to clearly explain when their algorithm is applicable.

[Author Response · NeurIPS 2020]

We thank the reviewers for their valuable feedback. We are glad that they found our approach "very interesting" (R4),
that it "sets a ... high bar for alternative[s]" (R2); our results "competitive" (R1) and "very promising" (R3); and our
experiments "refreshing"(R4). At the same time, the reviewers requested clarifications and improvements focused on the
novelty of our approach, and the discussion of key ideas. We respond in detail below to major questions/comments. A
final version of this paper will fully address all concerns raised by the reviewing team, including these major comments
and other minor suggestions.

**Novelty:** We address an "important and understudied problem in the [BO] literature" (R1), that "is very interesting
and can certainly find applications in important areas" (R4), using an "approach [that] brings together essentially all of
the ingredients of a real-world approach to [BO]" (R2). Although many of the ideas come from the existing literature,
"there is novelty in the combination of all of these ideas, which are numerous, ... and non-trivial." (R1).

Solving $\min_x \rho[F(x, W)]$ with BO, while leveraging the ability to choose $x$ *and* $w$ at query time is novel. As established
by Janusevskis & Le Roche (2013), and Toscano-Palmerin & Frazier (2018), the ability to select $w$ is crucial while
optimizing $\mathbb{E}[F(x, W)]$. Risk measures differ significantly from expectation, as the posterior risk-measure objective is
non-Gaussian and cannot be treated using a simple GP model. This introduces significant methodological challenges.
BO with risk measures, with ability to choose $w$, requires overcoming these challenges and provides significant value.

**Selecting $w$ is realistic and dramatically improves performance :** "select[ing] ... $w$ to query may not be realistic"
(R4). We left this implicit, but our setting is designed for making decisions about the real world using a *simulation*.
For example, in our COVID-19 experiment, $F(x, W)$ is the output of a simulation that models the real world. In the
simulation, we can set any $W$ we want. Then, once we find an approximate test allocation $x$, we would institute that in
the real world without knowing $W$. It is only at this point, after we choose a solution, that we cannot select $W$.

"the benchmark[s] ... select x, while you ... select both x and w"(R4). The existing literature cannot choose $w$ while
optimizing a risk measure. This and the corresponding statistical model are a major contribution of our paper and a
significant source of novelty, as evidenced by the superior performance of random sampling under our statistical model.

**Literature:** R1 points out two recent papers that we were not aware of. Gong, et al. (2019) is actually not relevant:
it considers parallel BO in a classical setting, and quantiles are used to induce diversity in the evaluation batches.
Torossian, et al. (2020) studies $\min_x \text{VaR}[F(x, W)]$. Our approach differs in that i) while they only pick $x$, we choose
$w$ as well, which is critical when $\mathcal{W}$ is large; ii) we allow for noisy observations of $F(x, w)$, which would introduce
additional bias in their method. We requested the code from the authors but they said that they were not yet ready to
release it, preventing inclusion of a comparison at the moment.

**Other comments:** "techniques to approximate VaR and CVaR ... seem to only hold for uniform distributions" (R4).
This is a simple misunderstanding, as we only use the uniform distribution for ease of exposition (see lines 142-5). In
fact, the distributions in two of our numerical experiments are non-uniform (Branin Williams and COVID).

"the estimators ... were studied in [42] and [43]" (R3). The estimators in [42] and [43] are for a given function $F$ where
the only randomness is in $W$, and require conditions on distribution of $F(\cdot, W)$. In our case $F$ is also random, and the
gradient is propagated through multiple operators (see §4 of supplement). This ties into "use ... [of] envelope theorem ...
popularized by Wu, et al. 2017" (R1) as the risk measures require more involved treatment than the posterior mean.

"QMC ... would offer faster convergence than simple [MC]" (R1). We use MC only for exposition. As discussed in
the supplement, we sample using (qMC) Sobol sequences in practice. "[G-H] quadrature for integrating over f" (R1):
The integrals are over $\rho$, which is non-Gaussian. "... "mild regularity" ... condition ... isn't continuity enough?" (R1)
The detailed conditions are discussed in §4 of supplement. They include differentiability of the GP sample paths and
conditions on the distribution of $W$.

"... experimental results ... comparing the run time..." (R4) An iteration of $\rho\text{KG}^{apx}$ takes between 1-3 minutes, running
on 4 cores of an Intel® Xeon® Platinum 8124M processor. The computational cost is justified by the sampling efficiency,
particularly when the function evaluations are expensive (hours). We will include relevant plots in the final version.

R1 and R4 ask for more discussion on the approximate method. $\rho\text{KG}^{apx}$ is inspired by the EI and KGCP algorithms,
which each consider only the previously evaluated points and show good numerical performance. A low-level intuition
is that the underlying GP model is an extrapolation of the data. Thus, these points carry an immense amount of
information on the GP model and the posterior objective, which makes them an ideal set of candidates to consider.

"... high-level discussion or motivation wrt alternative ideas" (R1) "The details ... in ... 5.1 and 5.2 ... difficult to follow
..." (R3) We will expand the discussion of key ideas, and clarify the details of our approach.

"LBFGS ... used ... gradient is ... with noise" (R1) We use SAA to trade the biased stochastic problem with a biased and
deterministic one. This allows the use of LBFGS which offers computational improvements over stochastic alternatives.

[Meta-Review · NeurIPS 2020]

Despite some initial concern about the novelty of the methodology, the reviewers were satisfied with the author response, and the overall importance of the problem. Please try to address the comments raised by the reviewers and promised in the response (especially regarding clarity of the algorithm).